# Short-Term CPAP Improves Biventricular Function in Patients with Moderate-Severe OSA and Cardiometabolic Comorbidities

**DOI:** 10.3390/diagnostics11050889

**Published:** 2021-05-17

**Authors:** Ioana Mădălina Zota, Radu Andy Sascău, Cristian Stătescu, Grigore Tinică, Maria Magdalena Leon Constantin, Mihai Roca, Daniela Boișteanu, Larisa Anghel, Ovidiu Mitu, Florin Mitu

**Affiliations:** 1Department of Medical Specialties (I), Faculty of Medicine, Grigore T. Popa—University of Medicine and Pharmacy, 700115 Iasi, Romania; madalina.chiorescu@gmail.com (I.M.Z.); leon_mariamagdalena@yahoo.com (M.M.L.C.); roca2m@gmail.com (M.R.); larisa.anghel@umfiasi.ro (L.A.); ovidiu@yahoo.co.uk (O.M.); mitu.florin@yahoo.com (F.M.); 2Department of Surgical Specialties (I), Faculty of Medicine, Grigore T. Popa—University of Medicine and Pharmacy, 700115 Iasi, Romania; grigoretinica@yahoo.com; 3Department of Medical Specialties (III), Faculty of Medicine, Grigore T. Popa—University of Medicine and Pharmacy, 700115 Iasi, Romania; boisteanu@yahoo.com

**Keywords:** OSA, CPAP, speckle tracking, LV-GLS

## Abstract

Obstructive sleep apnea (OSA) is the most common form of sleep-disordered breathing, exhibiting an increasing prevalence and several cardiovascular complications. Continuous positive airway pressure (CPAP) is the gold-standard treatment for moderate-severe OSA, but it is associated with poor patient adherence. We performed a prospective study that included 57 patients with newly diagnosed moderate-severe OSA, prior to CPAP initiation. The objective of our study was to assess the impact of short-term CPAP on ventricular function in patients with moderate-severe OSA and cardiometabolic comorbidities. The patients underwent a clinical exam, ambulatory blood pressure monitoring and comprehensive echocardiographic assessment at baseline and after 8 weeks of CPAP. Hypertension, obesity and diabetes were highly prevalent among patients with moderate-severe OSA. Baseline echocardiographic parameters did not significantly differ between patients with moderate and severe OSA. Short-term CPAP improved left ventricular global longitudinal strain (LV-GLS), isovolumetric relaxation time, transmitral E wave amplitude, transmitral E/A ratio, right ventricular (RV) diameter, RV wall thickness, RV systolic excursion velocity (RV S‘) and tricuspid annular plane systolic excursion (TAPSE). Short-term CPAP improves biventricular function, especially the LV-GLS, which is a more sensitive marker of CPAP-induced changes in LV systolic function, compared to LVEF. All these benefits are dependent on CPAP adherence.

## 1. Introduction

Obstructive sleep apnea (OSA) is the most common form of sleep-disordered breathing, characterized by repetitive upper airway collapse, despite normal ventilatory efforts [1,2]. Although primarily a sleep-related disorder, OSA promotes a cascade of neuro-hormonal changes lead by the activation of the renin-angiotensin aldosterone system, autonomic nervous function imbalance, oxidative stress and inflammation [1,2,3,4], and associated with increased cardiovascular risk [1,5,6].

Repetitive hypoxia increases afterload, promoting left ventricular (LV) diastolic dysfunction, which, in time, leads to subsequent impairment of the LV systolic function, especially in the presence of a chronic imbalance between myocardial oxygen demand and supply [7,8,9,10]. The use of conventional Doppler and 2D echocardiography in assessing diastolic function is limited by interference with patient age, heart rate and blood pressure values [9]. However, modern imaging techniques such as tissue doppler and speckle tracking echocardiography have enabled a more in-depth analysis of cardiac function in OSA patients. Speckle tracking ultrasonography (ST) differentiates active myocardial movement from tethering forces and is independent of the insonation angle [9]. The LV global longitudinal strain (LV-GLS) is the most studied ST parameter, characterized by high reproducibility [10] and higher sensitivity in detecting subclinical LV systolic dysfunction, compared to biplane Simpson’s ejection fraction (EF) [11,12,13,14,15].

Continuous positive airway treatment (CPAP)—the gold standard treatment option for moderate-severe OSA—maintains upper airway patency, effectively reducing the apnea hypopnea index (AHI). However, the benefits of CPAP are limited by poor patient adherence. The study of CPAP’s effect on LV systolic and diastolic function has yielded divergent results [13,14,15,16,17]. Although a recent meta-analysis of randomized control trials did not find significant changes in LVEF after CPAP [18], other studies have documented an improvement in LV-GLS after 12–24 weeks of CPAP [13,19].

The purpose of this study was to determine whether short-term CPAP is able to improve ventricular function in patients with moderate-severe OSA and cardiometabolic comorbidities.

## 2. Materials and Methods

We performed a prospective analysis that included 57 patients with newly diagnosed moderate-severe obstructive sleep apnea (prior to the initiation of CPAP therapy) and at least one of the following conditions: obesity, type 2 diabetes mellitus, impaired glucose tolerance, dyslipidemia or hypertension, or being admitted in our cardiovascular rehabilitation clinic between May 2018 and December 2018. Exclusion criteria were major surgery or acute medical conditions in the prior 30 days, use of supplemental oxygen, prior CPAP therapy, central sleep apnea, congenital heart disease, severe valvular heart disease, class IV NYHA heart failure and alcohol dependence.

All patients underwent a detailed medical history, clinical examination, ambulatory blood pressure monitoring (ABPM) and comprehensive transthoracic echocardiography before and after 2 months of CPAP. Obesity was defined as a body mass index (BMI) ≥ 30 kg/m^2^ and was classified as follows: grade 1 (BMI 30–34.9 kg/m^2^), grade 2 (BMI 35–40 kg/m^2^) and grade 3 (≥40 kg/m^2^). Hypertension was defined an office systolic blood pressure (SBP) ≥ 140 mmHg and/or diastolic blood pressure (DBP) ≥ 90 mmHg and was classified as follows: grade 1 (SBP 140–150 mmHg and/or DBP 90–99 mmHg), grade 2 (SBP 160–179 mmHg and/or DBP 100–110 mmHg) and grade 3 (SBP ≥ 180 mmHg and/or DBP ≥ 110 mmHg).

### 2.1. Sleep Study

OSA diagnosis was established in the local pneumology clinic by ambulatory or in-hospital six-channel cardiorespiratory polygraphy (Philips Respironics Alice Night One or DeVilbiss Porti 7). All recordings were manually scored by a trained pneumologist, according to the guidelines of the American Academy of Sleep Medicine (AASM). Apnea was defined as a complete cessation of airflow for a minimum of 10 s. Hypopnea was defined as a >50% reduction in airflow for a minimum of 10 s, associated with a ≥3% decrease in peripheral oxygen saturation or an arousal. Patients with AHI of 15–30 and >30 were considered to have moderate and severe OSA, respectively. CPAP effective pressure autotitration was performed in the sleep laboratory using Philips Respironics DreamStation Auto CPAP or a Resmed Airsense 10 Autoset.

### 2.2. Echocardiography

Standardized transthoracic echocardiography (2D, Doppler, TDI and speckle tracking) was performed by an experienced cardiologist according to EACVI guidelines [20] using PHILIPS EPIQ 5G. The first echocardiography was performed prior to the initiation of CPAP therapy. The second echocardiographic evaluation was performed after 8 weeks of CPAP (after a full night of CPAP use). All echocardiographies were performed on the same PHILIPS EPIQ 5G device, by the same operator.

2D and M-mode measurements (left ventricular end-diastolic diameter—LVEDD, interventricular septum—IVS, posterior wall thickness—PWT, end-diastolic diameter—RV) were performed in parasternal long axis (PLAX) with the patient in the left lateral decubitus. Right ventricular wall thickness was measured in diastole by M-mode or 2D imaging, preferably from the subcostal window.

Tricuspid and mitral annular plane systolic excursions (TAPSE and MAPSE) were calculated by M-mode thorough the tricuspid and mitral annulus in standard apical chamber view, respectively. Atrial volumes were calculated in telesystole by planimetry, excluding the area between the valvular leaflets and the annulus.

### 2.3. Doppler Analysis

Doppler-derived LV diastolic parameters (E and A peak velocities) were recorded by placing a pulsed wave cursor at the tips of the mitral valve. Isovolumic relaxation time (IVRT) was measured between the end of aortic ejection and the onset of mitral E wave velocity, with the PW Doppler sample volume in between the LV inflow and outflow tracts. Tissue doppler-derived parameters for the systolic and diastolic ventricular function (RV S’, lateral and medial LV S’, e’ and a’) were obtained in the mitral and tricuspid annulus from the apical 4-chamber view, ensuring optimal image orientation.

### 2.4. Global LV Longitudinal Strain Imaging

GLS measurements were performed with electrocardiographic monitoring in apical 4- 2- and 3-chamber views, adjusted for optimal spatial resolution (mitral annulus at the bottom of the image). The frame rate was set between 50 and 90 fps. The tracking of the regions of interest was performed offline using the PHILLIPS EPIQ 5G integrated software. Patients with poor image quality, arrhythmia and regional wall motion anomalies were excluded from strain imaging. Consequently, the effect of CPAP on LV GLS was analyzed in a subgroup of only 21 patients.

### 2.5. Ambulatory Blood Pressure Monitoring (ABPM)

The ABPM monitoring was performed with DMS-300 ABP (DM software) and was interpreted by a certified cardiologist. The frequency of daytime (06.00–22.00) and nighttime (22.00–06.00) blood pressure measurements was set at 30 and 60 min, respectively. The ABPM recording was deemed satisfactory if it included >70% of expected blood pressure measurements. The first ABPM recording was performed before the initiation of CPAP therapy. The second ABPM was performed after a full night of CPAP use.

### 2.6. CPAP Therapy

All patients were instructed regarding the importance of daily CPAP use and received medical advice promoting a healthy lifestyle, focusing on diet and exercise, without altering the preexisting chronic drug regimen. Patients received a standard 8 weeks of CPAP therapy with REMstar Auto C-Flex CPAP (Philips Respironics), DreamStation Auto CPAP (Philips Respironics) or AirSense 10 AutoSet CPAP (ResMed). Compliance to device therapy was defined as an average CPAP usage time ≥4 h/night, while non-compliance was defined as an average device usage time <4 h/night [21]. CPAP compliance data (device usage/hours/night at the prescribed pressure) was automatically recorded by the CPAP device and downloaded using the appropriate software (EncoreBasic v.2.1, Encore Pro 2 v.2.17 or ResScan v.6.0).

All patients signed a written informed consent for inclusion. The study was conducted in accordance with the Declaration of Helsinki, and the protocol was approved by the Ethics Committee of the “Grigore T. Popa” University of Medicine and Pharmacy in Iași.

Statistical analysis was performed in SPSS v 20.0. Descriptive data were expressed as means ± SD (standard deviation) or percentages. Chi-square and student’s t-test were used for comparisons between various groups. The Pearson correlation coefficient was used to evaluate a potential relationship between variables. A *p* value < 0.05 was considered statistically significant.

## 3. Results

Our study included 57 patients aged 36–79 years old with newly diagnosed moderate-severe OSA (AHI 42.3 ± 21.3 events/h) prior to the initiation of CPAP therapy. Demographic and clinical characteristics of our study population are illustrated in Table 1. The prevalence of different cardiometabolic comorbidities is depicted in Figure 1.

The prevalence of moderate and severe OSA was 38.59% and 61.40%, with an average AHI of 21.9 ± 4.04 and 55.11 ± 17.29 events/h, respectively. Table 2 illustrates the baseline echocardiographic parameters in our study group. Age, BMI and abdominal circumference did not vary between the two OSA severity subgroups (*p* > 0.05). AHI was not significantly correlated with age, weight, LVEF, LV s’, RV s’ or LV-GSL (*p* > 0.05). Systolic dysfunction (LVEF <50%) was present in seven cases (12.28%).

Overall, 42.10% of patients had an optimal CPAP adherence (>240 min/night). The average CPAP used in our study group (n = 57) was 243.12 min/night and 258.25 min/night in the subgroup that underwent LV-GLS strain analysis (n = 21). Information regarding CPAP adherence was not available in nine patients. Table 3 illustrates the short-term impact of CPAP therapy in moderate-severe OSA patients, with secondary analysis for CPAP adherent and non-adherent subgroups. Although LVEF, medial and lateral mitral S’ and MAPSE were not influenced by CPAP (*p* > 0.05), we documented a significant improvement in LV-GLS (2.57%, *p* = 0.001), which remained significant only in CPAP-adherent patients (2.95%, *p* = 0.007). Figure 2 depicts improvement in LV-GLS (Bulls Eye plot pattern) after 8 weeks CPAP in one of our patients. Right ventricular systolic function (TAPSE, RV-S’) significantly improved after CPAP, but the increase in TAPSE remained statistically significant only in the CPAP-adherent subgroup.

## 4. Discussion

The most important result of our prospective analysis is that short-term CPAP positively influences biventricular function in patients with moderate-severe OSA and cardiometabolic comorbidities, as shown by the improvement of LV-GLS, RV-S’ and TAPSE. Although LV-EF and MAPSE did not improve after CPAP, the augmentation of LV systolic function is supported by LV-GLS improvement, a more sensitive marker of LV systolic function.

With the exception of the left atrial volume, all baseline echocardiographic parameters had similar values in our moderate and severe OSA subgroups. This is in line with previous reports that identified statistically significant differences in echocardiographic parameters only between moderate-severe OSA patients and subjects with AHI <15 events/h [9]. Although it was previously shown that LA volume and area increase with OSA severity [9], LA volume was surprisingly higher in our moderate OSA subgroup. This paradoxical difference is due to the relatively low number of subjects in our study and the influence of associated cardiometabolic comorbidities.

Despite the known association between OSA and LV hypertrophy [15,22], the effect of CPAP on LV and LA structural parameters is controversial [13,14,15,16,17]. The lack of significant changes in LA volume or LV structural parameters in our study can be explained by the short duration of CPAP and by the lack of improvement in average 24-h blood pressure values.

E wave deceleration time (EDT) and isovolumetric relaxation time (IVRT) are higher in the presence of OSA (compared to non-OSA patients), but are not influenced by OSA severity [23,24,25,26,27], which is consistent with our findings. Contrary to our results, previous studies found that TDI parameters (E/e’, e’, a’, e’/a’) significantly vary with OSA severity [13,28]. However, the negative impact of OSA on LV diastolic function was indirectly shown in our analysis by the improvement in E, E/A and IVRT after short-term CPAP.

The effect of OSA on left ventricular systolic function has been extensively studied, but with divergent results [11,16,17,18,19,20]. LVEF is slightly lower in patients with moderate-severe OSA (Δ = −1.7%) [29]. LV-GLS is a more sensitive parameter of LV systolic function, exhibiting a circadian variation in OSA patients [22]. The average LV-GLS in our study group was −16.07 ± 3.81%, similar to that reported by Vural et al. [13], and close to the accepted pathological cut-off value (−16%) [30]. Although LV-GLS is significantly impaired in patients with severe OSA, compared to individuals with AHI <15 events/h [12,13,26], it had similar values in our moderate and severe OSA subgroups. Contrary to previous results [15,31], in our study, AHI was not correlated with LVEF or LV-GLS values.

Two previously published papers documented a significant benefit in LVEF (5%–23.9%) in patients with OSA and heart failure [14,32]. However, in agreement with our results, most studies [15,16] did not find significant changes in LVEF after CPAP [18]. On the other hand, evidence regarding the beneficial effect of CPAP on LV-GLS seems to be more consistent [13,19,22]. Our study documents a significant improvement in LV-GLS (Δ = 2.57%, *p* = 0.001), which remained significant only among patients with optimal adherence to CPAP (Δ = 2.95%, *p* = 0.007).

Patients with OSA are frequently associated with right ventricular failure [9] and have higher RA volumes [25]. Although RV S‘ was reported to be lower in patients with severe OSA compared to subjects with mild-moderate sleep apnea [33], in our study, RA and RV structural and functional parameters had similar values in the two apnea severity subgroups.

Dursunoglu et al. [34] reported no improvement in RV diameter after CPAP. However, our analysis confirms the results of Colish et al. [16], illustrating a decrease in RV diameter, which remained significant only in CPAP-adherent patients. In line with previous reports [14,28], CPAP also improved RV function (TAPSE and RV-S’).

Inconsistencies regarding the effectiveness of CPAP in reversing myocardial disfunction suggest that the effect of continuous positive airway pressure is vastly dependent on the duration of OSA (prior to diagnosis), patient comorbidities, CPAP adherence and duration. As an example, CPAP seems to be more effective in patients with dilated cardiomyopathy compared to chronic ischemic cardiomyopathy [35]. Although LVEF and MAPSE did not improve after CPAP, the augmentation of LV systolic function is supported by LV-GLS improvement. PSAP did not significantly decrease after CPAP, but RV function improvement was reflected by an increase in RV-S’ and TAPSE, as well by the decrease in RV diameter and RV wall thickness. Therefore, the correct assessment of cardiac structural and functional changes after CPAP in a complex patient with OSA and associated cardiometabolic comorbidities require the analysis of multiple parameters and individual interpretations of the results in specific clinical settings. Another important finding is that the improvement of some echocardiographic parameters (IVRT, LV-GLS, RV diameter, TAPSE, RV wall thickness) remained significant only among CPAP-adherent patients (>240 min/night), which underlines the importance of correct CPAP use.

Our patients also exhibited a mild, but statistically significant improvement in all analyzed anthropometric parameters (BMI, weight, abdominal circumference) after CPAP. However, subgroup analysis showed that the improvement in BMI remained significant only among CPAP-adherent patients, showing the importance of patient adherence in CPAP-facilitated weight loss.

The main limitations of our study are the absence of a control group and the relatively short follow-up duration (8 weeks). Our echocardiographic assessments were biased by the associated cardiometabolic comorbidities. However, it is our opinion that this reflects the benefits of CPAP in most OSA patients, who present multiple complex pathologies and poor CPAP adherence, needing a multidisciplinary approach.

## 5. Conclusions

Our study demonstrated that short-term CPAP improved biventricular function in patients with moderate-severe OSA and cardiometabolic comorbidities. This is reflected in the improvement of LV-GLS (a more sensitive marker of LV systolic function, compared to LVEF), RV-S’ and TAPSE. These changes indirectly suggest that OSA negatively influences cardiac performance. CPAP adherence influences the treatment outcome.

## Figures and Tables

**Figure 1 diagnostics-11-00889-f001:**
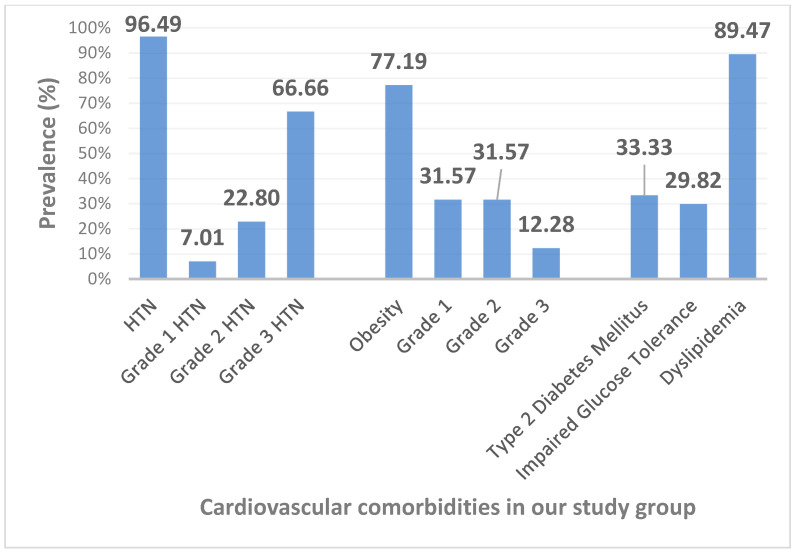
Prevalence of cardiometabolic comorbidities in patients with moderate-severe OSA. OSA—obstructive sleep apnea; HTN—hypertension.

**Figure 2 diagnostics-11-00889-f002:**
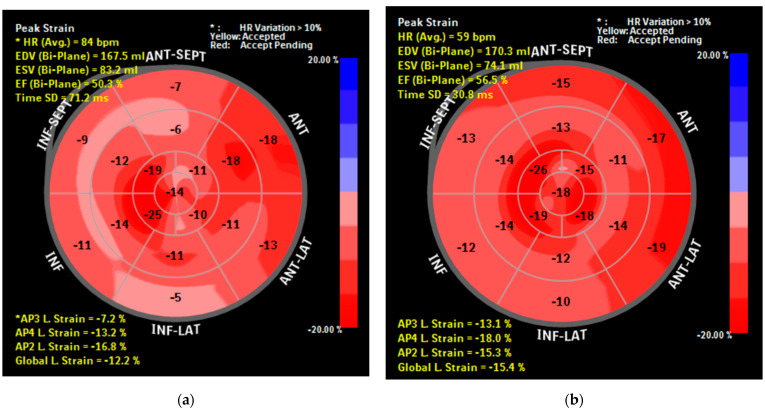
(**a**) Bull’s Eye diagram of LV-GLS before CPAP (patient G.C.); (**b**) Bull’s Eye diagram of LV-GLS after 8 weeks of CPAP (patient G.C.).

**Table 1 diagnostics-11-00889-t001:** Demographic and clinical characteristics of our study group.

Variable	Baseline Value
Age (years)	57.98 ± 9.17
Sex (m/f)	40/17
Weight (kg)	101.21 ± 17.63
BMI (kg/m^2^)	34.37 ± 5.99
Waist circumference (cm)	114.79 ± 11.43
Systolic Blood Pressure (mmHg)	137.21 ± 18.38
Diastolic Blood Pressure (mmHg)	86.47 ± 11.87
Systemic Hypertension (n)	55
Grade 1 (n)	4
Grade 2 (n)	13
Grade 3 (n)	38
Obesity (n)	44
Grade 1 (n)	18
Grade 2 (n)	18
Grade 3 (n)	7
Diabetes Mellitus (n)	19
Impaired glucose tolerance (n)	17
Dyslipidemia (n)	51
AHI (events/h)	42.3 ± 21.3
Desaturation index (events/h)	40.34 ± 20.86
Minimum nocturnal O2Sa (%)	71.58 ± 11.61
Average nocturnal O2Sa (%)	91.05 ± 3.72
CPAP pressure (cmH20)	11.31 ± 2.45

BMI = body mass index; AHI = apnea hypopnea index; O2Sa = oxygen saturation; CPAP = continuous positive airway pressure.

**Table 2 diagnostics-11-00889-t002:** Baseline echocardiographic parameters in our study group.

Variable	Baseline Value All Patients (n = 57)	Moderate OSA Subgroup (n = 22)	Severe OSA Subgroup (n = 35)	*p* *
Septal wall thickness (mm)	11.64 ± 1.76	11.51 ± 1.97	11.72 ± 1.64	NS
LV posterior wall thickness (mm)	11.86 ± 1.54	11.86 ±1.41	11.86 ± 1.64	NS
LV end-diastolic diameter (mm)	54.45 ± 7.31	54.01 ± 6.55	54.73 ± 7.83	NS
LV EF (%)	56.69 ± 6.81	57.74 ± 5.80	56.03 ± 7.37	NS
MAPSE (mm)	15.8 ± 3.50	15.47 ± 2.88	16 ± 3.87	NS
RV diameter (mm)	34 ± 3.97	33.58 ± 4.11	34.26 ± 3.92	NS
RV wall thickness (mm)	5.59 ± 1.25	5.44 ± 1.34	5.66 ± 1.19	NS
TAPSE (mm)	26.23 ± 5.60	26.18 ± 6.02	26.27 ± 5.42	NS
LA volume (ml)	75.22 ± 23.19	84.86 ± 29.79	69.28 ± 15.60	0.01
RA volume (ml)	56.23 ± 16.41	59.45 ± 18.74	54.77 ± 14.87	NS
Transmitral E wave (cm/s)	73.32 ± 20.50	71.26 ± 20.92	74.62 ± 20.43	NS
Transmitral A wave (cm/s)	74.88 ± 22.98	69.07 ± 26.10	78.68 ± 19.66	NS
Transmitral E/A ratio	1.04 ± 0.47	1.16 ± 0.61	0.97 ± 0.32	NS
Deceleration time (ms)	204.81 ± 64.09	220.90 ± 76.52	200 ± 69.23	NS
Isovolumetric relaxation time (ms)	117.54 ± 21.15	115.27 ± 25.19	115.42 ± 21.30	NS
Medial mitral e’ (cm/s)	7.87 ± 2.15	7.97 ± 1.73	7.77 ± 2.38	NS
Lateral mitral e’ (cm/s)	10.96 ± 3.33	10.71 ± 3.06	10.97 ± 3.60	NS
Medial mitral S’ (cm/s)	8.60 ± 1.71	8.17 ± 1.75	8.73 ± 1.76	NS
Lateral mitral S’ (cm/s)	9.29 ± 2.17	9.41 ± 2.43	9.09 ± 2.14	NS
E/average e’	7.97 ± 2.42	7.92 ± 3.26	8.33 ± 2.60	NS
RV S’ (cm/s)	13.47 ± 3.11	13.28 ± 3.59	13.96 ± 3.26	NS
PASP (mmHg)	21.97 ± 12.29	23.38 ± 13.32	21.12 ± 11.76	NS
LV-GLS (%)	−16.07 ± 3.81	−16.92 ± 3.70	−15.54 ± 3.93	NS

* = comparison between moderate and severe OSA subgroups; OSA = obstructive sleep apnea; LV = left ventricle; EF = ejection fraction; MAPSE = mitral annular plane systolic excursion; TAPSE = tricuspid annular plane systolic excursion; RV = right ventricle; LA = left atrium; RA = right atrium; PASP = pulmonary artery systolic pressure; LV-GLS = left ventricular global longitudinal strain.

**Table 3 diagnostics-11-00889-t003:** Comparison of clinical and echocardiographic parameters in our study group, before and after 8 weeks of CPAP.

	All Patients (n = 57)			CPAP Use < 240 Min/Night (n = 24)			CPAP Use ≥ 240 Min/Night (n = 24)		
Variable	Baseline	After 8 Weeks CPAP	*p* *	Baseline	After 8 Weeks CPAP	*p* ”	Baseline	After 8 Weeks CPAP	*p* #
Weight (kg)	101.21	99.07	0.000002	109.16	106.31	0.0003	95.77	94.14	0.01
BMI (kg/m^2^)	34.36	33.83	0.01	36.56	35.88	NS	33.37	32.80	0.04
Waist circumference (cm)	114.78	111.97	0.000001	119.79	115.77	0.00006	111.62	109.20	0.002
SBP/24 h	130.45	130.89	NS	129.25	130.20	NS	130.00	130.70	NS
DBP/24 h	75.66	74.87	NS	76.62	75.04	NS	74.16	74.29	NS
Septal wall thickness (mm)	11.64	11.41	NS	11.52	11.4	NS	11.75	11.33	NS
LV posterior wall thickness (mm)	11.86	11.65	NS	11.57	11.49	NS	11.87	11.53	NS
LV end-diastolic diameter (mm)	54.45	53.27	NS	54.06	52.17	NS	54.97	53.75	NS
LV EF (%)	56.69	56.49	NS	53.78	53.97	NS	59.42	58.37	NS
MAPSE (mm)	15.8	16.7	NS	15.6	15.85	NS	15.98	17.06	NS
RV diameter (mm)	33.83	32.97	0.01	34.39	33.31	NS	32.34	31.35	0.04
RV wall thickness (mm)	5.59	5.20	0.008	5.63	5.32	NS	5.46	4.96	0.02
TAPSE (mm)	26.23	27.92	0.03	26.68	27.67	NS	25.81	29.04	0.007
LA volume (mL)	75.22	76.43	NS	72.45	72.69	NS	76.47	79.36	NS
RA volume (mL)	56.23	56.95	NS	54.39	54.57	NS	56.77	57.91	NS
Transmitral E wave (cm/s)	73.32	81.69	0.0002	69.32	79.49	0.007	74.01	83.33	0.01
Transmitral A wave (cm/s)	74.88	74.84	NS	70.86	69.69	NS	77.6	81.44	NS
Transmitral E/A ratio	1.04	1.15	0.02	1.02	1.16	0.01	1.02	1.10	NS
Deceleration time (ms)	204.81	216.65	NS	203.16	215	NS	196.22	216.04	NS
Isovolumetric relaxation time (ms)	117.54	109.69	0.01	114.41	110.83	NS	118.22	108	0.02
Medial mitral e’ (cm/s)	7.87	7.95	NS	8.17	7.47	NS	7.82	8.05	NS
Lateral mitral e’ (cm/s)	10.96	11.16	NS	12	11.70	NS	10.09	10.56	NS
Medial mitral S’ (cm/s)	8.60	8.52	NS	9.50	9.61	NS	8.28	8.34	NS
Lateral mitral S’ (cm/s)	9.29	9.46	NS	8.45	8.45	NS	8.67	9.24	NS
E/average e’	7.97	8.71	NS	7.32	8.47	NS	8.11	9.06	NS
RV S’ (cm/s)	13.47	14.34	0.01	13.35	14.66	0.01	13.32	14.29	0.02
PASP (mmHg)	21.97	21.14	NS	20.22	17.20	NS	24.34	26.22	NS
LV-GLS (%)	−16.07	−18.64	0.001	−15.6	−18.56	NS	−15.15	−18.1	0.007

* = comparison between variables at baseline and after 8 weeks of CPAP in all patients; ” = comparison between variables at baseline and after 8 weeks of CPAP in non-compliant patients; # = comparison between variables at baseline and after 8 weeks of CPAP in CPAP-compliant patients; OSA = obstructive sleep apnea; LV = left ventricle; EF = ejection fraction; MAPSE = mitral annular plane systolic excursion; TAPSE = tricuspid annular plane systolic excursion; RV = right ventricle; LA = left atrium; RA = right atrium; PASP = pulmonary artery systolic pressure; LV-GLS = left ventricular global longitudinal strain.

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
