# Peer review of "Short-Term CPAP Improves Biventricular Function in Patients with Moderate-Severe OSA and Cardiometabolic Comorbidities"

_diagnostics, 2021, doi:10.3390/diagnostics11050889_

Round 1
Reviewer 1 Report
The authors did an excellent job in addressing this important area. Though the results for the EF and TAPSE are not significant LV-GLS was significant.
Would double check as line 92 states that the second echo was done after ist night of cpap. Why any significance. Was the echo done again on the 8th week? Please clarify?
Define the grades of hypertension and obesity in the text
Author Response
We sincerely thank the reviewer for his/her thoughtful comments and constructive suggestions. We have revised our manuscript in light of these recommendations and completed our manuscript with new information, which hopefully improve the quality of our paper.
- Would double check as line 92 states that the second echo was done after ist night of cpap. Why any significance. Was the echo done again on the 8th week? Please clarify?
The persistence of CPAP beneficial effects after CPAP withdrawal is currently under debate. We wanted to emphasize that the second echocardiography was performed after 8 weeks of CPAP and that the patient had used the device during the previous night. We have rephrased the paragraph, hoping that the methodology is more clear now (lines 108-109).
” The second echocardiographic evaluation was performed after 8 weeks of CPAP (after a full night of CPAP use).”
2.Define the grades of hypertension and obesity in the text
We have added the required information in the manuscript (lines 86-92).
” Obesity was defined as a body mass index (BMI) ≥ 30 kg/m2, and was classified as follows: grade 1 (BMI 30-34.9 kg/m2), grade 2 (BMI 35-40 kg/m2) and grade 3 (≥40 kg/m2). Hy-pertension was defined an office systolic blood pressure (SBP) ≥ 140 mmHg and/or diastolic blood pressure (DBP) ≥ 90 mmHg and was classified as follows: grade 1 (SBP 140-150 mmHg and/or DBP 90-99 mmHg), grade 2 (SBP 160-179 mmHg and/or DBP 100-110 mmHg) and grade 3 (SBP ≥180 mmHg and/or DBP ≥110 mmHg).”
We hope that you find our responses satisfactory and that our revised manuscript is now suitable for publication,
Sincerely,
All authors
Reviewer 2 Report
The study entitled “Short term CPAP improves biventricular function in patients with moderate-severe OSA and cardio-metabolic comorbidities “, is interesting and well-presented. The authors aimed to assess the impact of short-term CPAP on ventricular function in patients with moderate-severe OSA and cardio-metabolic comorbidities. The authors concluded short-term CPAP therapy improves biventricular function, especially the LV-GLS.
Congratulations to the authors for completing this study of high clinical importance. Besides, I have highlighted several points in which this manuscript needs to be improved.
- In the abstract, the authors did not write out the full name of RV S` when it first appeared, please add.
- Figure 1, it is recommended that the author mark “%” on the Y axis instead of marking % on the top of the bar graph to make the picture more concise.
- In the table 2, in order for readers to read better, please add patient number (n) of the moderate OSA, and severe OSA subgroups.
- In the table 3, the entire table caption is missing, please add it.
- The discussion section is too lengthy. Taking this paper as an example, the authors should concisely integrate the past literature, and then focus on explaining the findings and significance of this research and how to apply it in the clinic. Please modify and reduce the length of the discussion.
- In the last paragraph of discussion, the authors stated “it is our opinion that this reflects the benefits of CPAP in the ”real world OSA patient”, with multiple complex pathologies and poor CPAP adherence, who needs a multidisciplinary approach.”. However, real-world evidence (RWE) in medicine means evidence obtained from real world data, which are generated during routine clinical practice. This study is a prospective study and does not truly fit the definition of RWE. Therefore, please correct the statement and delete “real world”.
Author Response
We sincerely thank the reviewer for his/her thoughtful comments and constructive suggestions. We have revised our manuscript in light of these recommendations and completed our manuscript with new information, which hopefully improve the quality of our paper.
- In the abstract, the authors did not write out the full name of RV S` when it first appeared, please add.
We rephrased the sentence according to the reviewer’s suggestions (lines 28-29).
”Short-term CPAP improved left ventricular global longitudinal strain (LV-GLS), isovolumetric relaxation time, transmitral E wave amplitude, transmitral E/A ratio, right ventricular (RV) diameter, RV wall thickness, RV systolic excursion velocity (RV S`) and tricuspid annular plane systolic excursion (TAPSE).”
- Figure 1, it is recommended that the author mark “%” on the Y axis instead of marking % on the top of the bar graph to make the picture more concise.
We modified the figure according to the reviewer’s suggestions (line 173).
- In the table 2, in order for readers to read better, please add patient number (n) of the moderate OSA, and severe OSA subgroups.
We added the number of patients in each subgroup.
|
Variable |
Baseline value All patients (n=57) |
Moderate OSA subgroup (n=22) |
Severe OSA subgroup (n=35) |
p* |
- In the table 3, the entire table caption is missing, please add it.
We added the missing table caption (line 202).
”Table 3. Comparision of clinical and echocardiographic parameters in our study group, before and after 8 weeks CPAP”
- The discussion section is too lengthy. Taking this paper as an example, the authors should concisely integrate the past literature, and then focus on explaining the findings and significance of this research and how to apply it in the clinic. Please modify and reduce the length of the discussion.
We have shortened the discussion section (lines 212-418).
” The most important result of our prospective analysis is that short term CPAP pos-itively influences biventricular function in patients with moderate-severe OSA and car-dio-metabolic comorbidities, as shown by the improvement of LV-GLS, RV-S’ and TAPSE. Although LV EF and MAPSE did not improve after CPAP, augmentation of LV systolic function is supported by LV-GLS improvement, a more sensitive marker of LV systolic function.
With the exception of left atrial volume, all baseline echocardiographic parameters had similar values in our moderate and severe OSA subgroups. This is in line with previous reports that identified statistically significant differences in echocardiographic parameters only between moderate-severe OSA patients and subjects with AHI<15 events/h [9]. Although it was previously shown that LA volume and area increase with OSA severity [9], LA volume was surprisingly higher in our moderate OSA subgroup. This paradoxical difference is due to the relatively low number of subjects in our study and the influence of associated cardiometabolic comorbidities.
Despite the known association between OSA and LV hypertrophy [15,28], the effect of CPAP on LV and LA structural parameters is controversial [13–17]. The lack of sig-nificant changes in LA volume or LV structural parameters in our study can be explained by the short duration of CPAP and by the lack of improvement in average 24h-blood pressure values.
E wave deceleration time (EDT) and isovolumetric relaxation time (IVRT) are higher in the presence of OSA (compared to non-OSA patients), but are not influenced by OSA severity [23–25], which is consistent with our findings. Contrary to our results, previous studies found that TDI parameters (E/e`, e`, a`, e`/a`) significantly vary with OSA sever-ity[13,34]. However, the negative impact of OSA on LV diastolic function is indirectly shown in our analysis by the improvement in E, E/A and IVRT after short-term CPAP.
The effect of OSA on left ventricular systolic function has been extensively studied, but with divergent results [11,16–20]. LVEF is slightly lower in patients with moder-ate-severe OSA (Δ= -1.7%)[27]. LV-GLS is a more sensitive parameter of LV systolic function, exhibiting a circadian variation in OSA patients [28]. The average LV-GLS in our study group was -16.07± 3.81%, similar to that reported by Vural et al.[13], and close to the accepted pathological cut-off value (-16%) [29]. Although LV-GLS is significantly impaired in patients with severe OSA, compared to individuals with AHI<15 events/h [12,13,25], it has similar values in our moderate and severe OSA subgroups. Contrary to previous results [15,30], in our study AHI was not correlated with LVEF or LV-GLS values.
Two previously published papers documented a significant benefit in LVEF (5-23.9%) in patients with OSA and heart failure[14,31]. However, in agreement with our results, most studies[15,16] did not find significant changes in LVEF after CPAP[18]. On the other hand, evidence regarding the beneficial effect of CPAP on LV-GLS seems to be more consistent [13,19,28]. Our study documents a significant improvement in LV-GLS (Δ=2.57%, p=0.001), which remained significant only among patients with optimal ad-herence to CPAP (Δ=2.95%, p=0.007).
Patients with OSA frequently associate right ventricular failure [9] and have higher RA volumes [24]. Although RV S` was reported to be lower in patients with severe OSA compared to subjects with mild-moderate sleep apnea [32], in our study, RA and RV structural and functional parameters had similar values in the 2 apnea severity sub-groups.
Dursunoglu et al.[33] reported no improvement in RV diameter after CPAP. Howevever, our analysis, confirms the results of Colish et al.[16], illustrating a decrease in RV diameter, which remained significant only in CPAP adherent patients. In line with previous reports [14,34], CPAP also improved RV function (TAPSE and RV-S’).
Inconsistencies regarding the effectiveness of CPAP in reversing myocardial dis-function suggest that the effect of continuous positive airway pressure is vastly dependent on the duration of OSA (prior to diagnosis), patient comorbidities, CPAP adherence and duration. As an example, CPAP seems to be more effective in patients with dilated cardiomyopathy compared to chronic ischemic cardiomyopathy [35]. Although LVEF and MAPSE did not improve after CPAP, augmentation of LV systolic function is supported by LV-GLS improvement. PSAP did not significantly decrease after CPAP, but RV function improvement is reflected by an increase in RV- S` and TAPSE, as well by the decrease in RV diameter and RV wall thickness. Therefore, the correct assessment of cardiac structural and functional changes after CPAP in a complex patient with OSA and associated car-dio-metabolic comorbidities requires the analysis of multiple parameters and individual interpretation of the results in specific clinical settings. Another important finding is that the improvement of some echocardiographic parameters (IVRT, LV-GLS, RV diameter, TAPSE, RV wall thickness) remained significant only among CPAP adherent patients (>240 minutes/night), which underlines the importance of correct CPAP use.
Our patients also exhibited a mild, but statistically significant improvement in all analyzed anthropometric parameters (BMI, weight, abdominal circumference) after CPAP. However, subgroup analysis showed that the improvement in BMI remained significant only among CPAP adherent patients, showing the importance of patient adherence in CPAP-facilitated weight loss.
The main limitations of our study are the absence of a control group and the relatively short follow-up duration (8 weeks). Our echocardiographic assessments were biased by the associated cardiometabolic comorbidities. However, it is our opinion that this reflects the benefits of CPAP in most OSA patients, who present multiple complex pathologies and poor CPAP adherence, needing a multidisciplinary approach”
- In the last paragraph of discussion, the authors stated “it is our opinion that this reflects the benefits of CPAP in the ”real world OSA patient”, with multiple complex pathologies and poor CPAP adherence, who needs a multidisciplinary approach.”. However, real-world evidence (RWE) in medicine means evidence obtained from real world data, which are generated during routine clinical practice. This study is a prospective study and does not truly fit the definition of RWE. Therefore, please correct the statement and delete “real world”.
We have rephrased the last paragraph (line 417).
” However, it is our opinion that this reflects the benefits of CPAP in most OSA patients, who present multiple complex pathologies and poor CPAP adherence, needing a multidisciplinary approach.”
We hope that you find our responses satisfactory and that our revised manuscript is now suitable for publication,
Sincerely,
All authors
Reviewer 3 Report
The study is correctly designed and the results are clearly and transparently presented. The set goals correspond to the conclusions.The findings are interesting and innovative, and underline the important aspect – an improvement of some echocardiographic parameters (IVRT, LV-GLS, RV diameter, TAPSE, RV wall thickness) after CPAP. All these benefits are dependent on CPAP adherence. It underlines the importance of correct CPAP use.
I have no comments for work.
I would like to recommend this manuscript for print in Journal MDPI – diagnstics.
Author Response
We thank the reviewer for his/her thoughtful comments and appreciation of our work.
Sincerely,
All authors